# HOCKEY-STICK GAN

**Edgar Minasyan**
Massachusetts Institute of Technology
Cambridge, MA 02139, USA
{minasyan}@mit.edu

**Vinay Uday Prabhu**
UnifyID
San Francisco, CA 94107, USA
{vinay}@unify.id

## ABSTRACT

We propose a new objective for generative adversarial networks (GANs) that is aimed to address current issues in GANs such as mode collapse and unstable convergence. Our approach stems from the hockey-stick divergence that has properties we claim to be of great importance in generative models. We provide theoretical support for the model and preliminary results on synthetic Gaussian data.

## 1 INTRODUCTION

Generative Adversarial Networks (GANs) (Goodfellow et al., 2014) has been an extremely successful class of generative models since its emergence. Due to its notoriously hard training and convergence issues as well as mode collapse (Goodfellow, 2016), a number of different versions of GANs have arisen (see Poole et al. (2016), Zhao et al. (2016), Chen et al. (2016), etc). The core idea behind all the GANs is a 2-player minimax game between the generator $G$ and the discriminator $D$. The generator tries to generate new samples $x \sim \mathbb{P}_g$ taking in noise $z \sim \mathbb{P}_z$ as input so that the generated distribution $\mathbb{P}_g \approx \mathbb{P}_r$ is close to the real distribution. The discriminator's goal is to differentiate between samples from $\mathbb{P}_g$ and $\mathbb{P}_r$ in order to make $G$ strive for indistinguishable results.

A modification in the objective of the minimax game changes the optimization of the model, potentially giving rise to a new GAN. A desired property of the objective is having the minimization part by the generator with an optimal (under constraints) discriminator indicate $\mathbb{P}_g$ converging to $\mathbb{P}_r$ in some sense. For instance, the original vanilla GAN has the generator minimizing the Jensen-Shannon divergence assuming an optimal discriminator. Similarly, Wasserstein-GAN (Arjovsky et al., 2017) was developed to have the generator minimize the Earth-Mover distance. In this paper, we present a new GAN objective (HS-GAN) that corresponds to the so called hockey-stick divergence and explore possible theoretical and practical properties of it. It is worth to take a look at the general version of a GAN objective before proceeding to the specific cases. The minimax game is represented as follows

$$\min_G \max_D \mathbb{E}_{x \sim \mathbb{P}_r} [m(D(x))] + \mathbb{E}_{x \sim \mathbb{P}_g} [m(1 - D(x))], \tag{1}$$

with an increasing and concave measuring function $m(\cdot)$ (Arora et al., 2017). Note that the objective (1) is consistent with the intuition described above as the discriminator aims to have a high score for real data and low score for generated data while the generator tries to fool the discriminator in believing generated data is real, i.e. high score for generated data. The remaining sections introduce the Hockey-Stick GAN and investigate its advantages and drawbacks both theoretically and practically.

## 2 HOCKEY-STICK DIVERGENCE

The hockey-stick divergence (Sason & Verdu, 2016) is a member of the $f$-divergence family.

**Definition 1.** The $f$-divergence family is a family of divergences between probability distributions given by

$$D_f(P \parallel Q) = \int_{\mathcal{X}} q(x) f\left(\frac{p(x)}{q(x)}\right) dx$$

where $f : \mathbb{R}^+ \to \mathbb{R}$ is a convex, lower semicontinuous function.

This is a very well-known family in information theory with a number of essential properties. It includes the famous Kullback-Leibler, Jensen-Shannon, $\chi^2$, total variation and other divergences. The hockey-stick divergence is an extension of the total variation distance.

**Definition 2.** The hockey-stick divergence is the $f$-divergence corresponding to the 'hockey-stick' function $f_\gamma(t) = \max(t - \gamma, 0)$ with $\gamma \geqslant 1$,

$$E_\gamma(P \parallel Q) = D_{f_\gamma}(P \parallel Q) = \int_{\mathcal{X}} q(x) \max\left(\frac{p(x)}{q(x)} - \gamma, 0\right) dx = \int_{p(x) \geqslant \gamma q(x)} (p(x) - \gamma q(x)) dx$$

Notice that when $\gamma = 1$, we have that the hockey-stick divergence $E_{\gamma=1}(P \parallel Q) = \frac{1}{2}|P - Q|$ is equivalent to the total variation distance.

There are several properties of the hockey-stick divergence that are worth to consider for the later analysis of HS-GAN. First, unlike most of the common divergences, $E_\gamma(P \parallel Q)$ with $\gamma > 1$ can equal zero even when $P$ and $Q$ are different (see Appendix). This is not desirable as one cannot guarantee anything certain about closeness of $P$ and $Q$ even when having zero divergence between them.

The most attractive part of the hockey-stick divergence is its flexibility of the parameter $\gamma$. According to (Sason & Verdu, 2016), it contains enough information to determine KL, Hellinger and almost any other $f$-divergence. More specifically, the set of values

$$\{(E_\gamma(P \parallel Q), E_\gamma(Q \parallel P)), \gamma \geqslant 1\}$$

is known to uniquely determine the values of all $f$-divergences with twice differentiable $f$. It is straightforward to show that $\{E_\gamma(Q \parallel P), \gamma \geqslant 1\}$ is equivalent to $\{E_\gamma(P \parallel Q), 0 < \gamma \leqslant 1\}$ (see Appendix). Hence, we conclude that the set of values $\{E_\gamma(P \parallel Q), \gamma > 0\}$ is enough to determine the values of almost all $f$-divergences. The information stored in the hockey-stick divergence with a flexible $\gamma$ can serve as a great advantage in the design of a generative model.

## 3 HS-GAN

### 3.1 METHOD OF $f$-GAN

The $f$-GAN method introduced by (Nowozin et al., 2016) is a general technique for the family of $f$-divergences. In particular, it describes a method for constructing objective functions that become equivalent to an $f$-divergence after optimizing the discriminator. The main result of the method is the following variational bound on $f$-divergences

$$D_f(P \parallel Q) \geqslant \sup_{T \in \mathcal{T}} \left( \mathop{\mathbb{E}}_{x \sim P}[T(x)] - \mathop{\mathbb{E}}_{x \sim Q}[f^*(T(x))] \right) \tag{2}$$

where $\mathcal{T}$ is an arbitrary class of functions $T : \mathcal{X} \to \mathbb{R}$ and $f^*(\cdot)$ is the *Fenchel conjugate* of $f(\cdot)$ defined as $f^*(t) = \sup_{u \in dom_f}\{ut - f(u)\}$. The bound (2) follows from the Fenchel-Moreau theorem that the biconjugate $f^{**} = f$ is the function itself when $f$ is convex and lower semicontinuous as well as Jensen's inequality to switch the order of integration and supremum. Fortunately, this bound is also tight, specifically for

$$T_{opt}(x) = f'\left(\frac{p(x)}{q(x)}\right)$$

### 3.2 HOCKEY-STICK GAN

Having the heavy artillery of $f$-GAN, we proceed to designing the Hockey-Stick GAN presented in the following theorem.

**Theorem 1.** *If the range of $D$ is $[0, 1]$, then*

$$\max_D \mathop{\mathbb{E}}_{x \sim \mathbb{P}_r}[D(x)] - \gamma \mathop{\mathbb{E}}_{x \sim \mathbb{P}_g}[D(x)] = E_\gamma(\mathbb{P}_r \parallel \mathbb{P}_g), \tag{3}$$

$$\mathbb{1}_{p_r(x) \geqslant \gamma p_g(x)} = \arg\max_D \mathop{\mathbb{E}}_{x \sim \mathbb{P}_r}[D(x)] - \gamma \mathop{\mathbb{E}}_{x \sim \mathbb{P}_g}[D(x)] \tag{4}$$

Proofs of this theorem can be found in the Appendix. Notice that (3) implies that HS-GAN with the objective

$$\min_G \ \max_{D:\mathcal{X}\to[0,1]} \ \mathbb{E}_{x\sim\mathbb{P}_r}\big[D(x)\big] - \gamma \ \mathbb{E}_{x\sim\mathbb{P}_g}\big[D(x)\big], \text{ where } \gamma > 0 \qquad (5)$$

has the generator minimizing the hockey-stick divergence in case of an optimal discriminator. This is the exact property of HS-GAN that makes the model worthwhile. Moreover, ensuring the range constraint of $[0, 1]$ on the discriminator has a standard solution to it such as having a *sigmoid* or *tanh* output activation function.

Even though theorem 1 provides a theoretical justification for HS-GAN, a substantial downside of HS-GANs comes from the nature of the hockey-stick divergence. As mentioned in section 2, $E_\gamma$ for $\gamma > 1$ does not satisfy the identity condition that the majority of divergences satisfy. Hence, there are no guarantees of closeness of distributions $P, Q$ even if $E_\gamma(P \parallel Q) = 0$. In terms of HS-GAN, this means that even though in presence of an optimal discriminator the generator is minimizing the hockey-stick divergence, there is no guarantee that $\mathbb{P}_g \approx \mathbb{P}_r$ even if the generator converges at a global minimum.

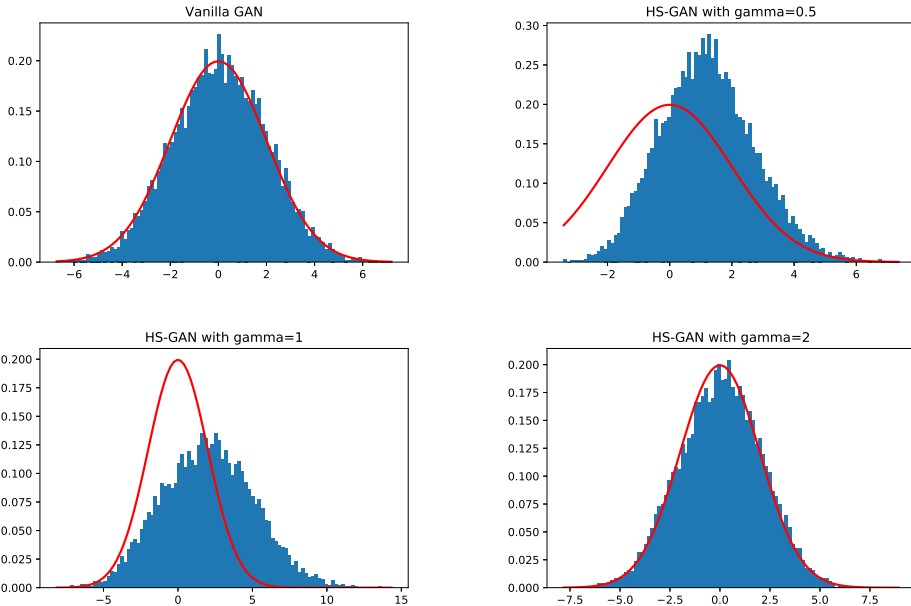

Figure 1: Experimental results on fitting a Gaussian distribution using vanilla GAN and Hockey-Stick GAN. *Red line* indicates the real (normal) distribution with mean $\mu = 0$ and variance $\sigma^2 = 2$. *Blue histograms* represent the generated distributions by vanilla GAN (*left top*) and HS-GAN with $\gamma = 0.5, 1, 2$. Notice that $\gamma = 2$ fits the real distribution even though others fail to do so.

## 4 EXPERIMENTS

In order to illustrate the properties of HS-GAN, we run simple experiments with low-capacity neural networks over synthetic Gaussian data (see Figure 1). The experiments show convergence of HS-GAN to different expected distributions while varying the value of gamma. The obtained distributions correspond to the intuition behind the properties of the hockey-stick divergence since, as mentioned earlier, distributions don't have to be identical for the hockey-stick divergence to equal 0. However, we can already see the usefulness of the varying gamma parameter that could be leveraged in a number of ways. Despite many other conducted experiments, more foundational practical results are left for future work.

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

APPENDIX

### 4.1 Hockey-Stick properties

**Claim 1.** *There exist non-identical distributions $P, Q$ such that $E_\gamma(P \parallel Q) = 0$ for all $\gamma > 1$.*

*Proof.* Consider the following Bernoulli distributions $P = Ber(\frac{1+\gamma}{2\gamma})$ and $Q = Ber(\frac{1}{\gamma})$ for $\gamma > 1$. Notice that

$$\frac{1+\gamma}{2\gamma} = p(0) < \gamma q(0) = 1, \ \frac{\gamma - 1}{2\gamma} = p(1) < \gamma q(1) = \gamma - 1$$

which means that $E_\gamma(P \parallel Q) = 0$ while $P \neq Q$. $\qquad \square$

**Claim 2.** *The two sets $\{E_\gamma(Q \parallel P), \gamma \geqslant 1\}$ and $\{E_\gamma(P \parallel Q), 0 < \gamma \leqslant 1\}$ are equivalent in terms of information stored.*

*Proof.* Take $0 < \gamma \leqslant 1$ and consider

$$E_{\frac{1}{\gamma}}(Q \parallel P) = \int_{q(x) \geqslant p(x)/\gamma} (q(x) - \frac{1}{\gamma}p(x))dx = \frac{1}{\gamma}\left(-\int_{\gamma q(x) > p(x)} (p(x) - \gamma q(x))dx\right) =$$

$$= \frac{1}{\gamma}\left(-\int (p(x) - \gamma q(x))dx + \int_{p(x) \geqslant \gamma q(x)} (p(x) - \gamma q(x))dx\right) = \frac{1}{\gamma}(\gamma - 1 + E_\gamma(P \parallel Q))$$

Thus, for each $0 < \gamma \leqslant 1$ the divergence values $E_{\frac{1}{\gamma}}(Q \parallel P)$ and $E_\gamma(P \parallel Q)$ contain equivalent information and the claim is proved. $\qquad \square$

### 4.2 Proofs

**Theorem.** *If the range of $D$ is $[0, 1]$, then*

$$\max_D \mathbb{E}_{x \sim \mathbb{P}_r}[D(x)] - \gamma \mathbb{E}_{x \sim \mathbb{P}_g}[D(x)] = E_\gamma(\mathbb{P}_r \parallel \mathbb{P}_g),$$

$$\mathbb{1}_{p_r(x) \geqslant \gamma p_g(x)} = \arg\max_D \mathbb{E}_{x \sim \mathbb{P}_r}[D(x)] - \gamma \mathbb{E}_{x \sim \mathbb{P}_g}[D(x)]$$

*Proof.* Let us first compute the Fenchel conjugate of $f_\gamma$ given by $f_\gamma^*(t) = \sup_{u \geqslant 0}\{ut - f_\gamma(u)\}$. When $u \geqslant \gamma$ we have $\sup_{u \geqslant \gamma}\{ut - u + \gamma\} = \gamma t$ for $t \leqslant 1$ and $\sup_{u \geqslant \gamma}\{ut - u + \gamma\} = +\infty$ for $t > 1$. Similarly, when $0 \leqslant u \leqslant \gamma$ we get that $\sup_{0 \leqslant u \leqslant \gamma}\{ut\} = \gamma t$ for $t \geqslant 0$ and $\sup_{0 \leqslant u \leqslant \gamma}\{ut\} = 0$ for $t < 0$. Hence, overall we get that

$$f_\gamma^*(t) = \begin{cases} +\infty, & \text{for } t > 1 \\ \gamma t, & \text{for } 0 \leqslant t \leqslant 1 \\ 0, & \text{for } t < 0 \end{cases}$$

Given the assumption that the range of $D$ is $[0, 1]$, we obtain from (2) that

$$E_\gamma(\mathbb{P}_r \parallel \mathbb{P}_g) = D_{f_\gamma}(\mathbb{P}_r \parallel \mathbb{P}_g) \geqslant \sup_{D: \mathcal{X} \to [0,1]}\left(\mathbb{E}_{x \sim \mathbb{P}_r}[D(x)] - \gamma \mathbb{E}_{x \sim \mathbb{P}_g}[D(x)]\right) \qquad (6)$$

In addition to (6), it is easy to check that $D_{opt}(x) = f_\gamma'(\frac{p_r(x)}{p_g(x)}) = \mathbb{1}_{p_r(x) \geqslant \gamma p_g(x)}$ satisfies equality under the constraint of $[0, 1]$ range

$$\mathbb{E}_{x \sim \mathbb{P}_r}[D_{opt}(x)] - \gamma \mathbb{E}_{x \sim \mathbb{P}_g}[D_{opt(x)}] = \int_\mathcal{X} (p_r(x) - \gamma p_g(x))\mathbb{1}_{p_r(x) \geqslant \gamma p_g(x)}dx = E_\gamma(\mathbb{P}_r \parallel \mathbb{P}_g)$$

which along with (6) shows both (3) and (4) and finishes the proof of the theorem.

$\qquad \square$

*Alternate Proof of Theorem.* Alternative to the $f$-GAN method, it is possible to provide an elementary proof to the theorem. To prove the statements (3) and (4) it is enough to just break up the integral and bound each separately

$$\underset{x \sim \mathbb{P}_r}{\mathbb{E}} [D(x)] - \gamma \underset{x \sim \mathbb{P}_g}{\mathbb{E}} [D(x)] = \int_{\mathcal{X}} (p_r(x) - \gamma p_g(x)) D(x) dx =$$

$$= \int_{p_r(x) \geqslant \gamma p_g(x)} (p_r(x) - \gamma p_g(x)) D(x) dx + \int_{p_r(x) < \gamma p_g(x)} (p_r(x) - \gamma p_g(x)) D(x) dx$$

To upper bound the first integral notice that as $p_r(x) - \gamma p_g(x) \geqslant 0$ we can simply replace $D(x)$ by its upper bound, i.e. $1$. The second integral is less than or equal to $0$ as for each $x$ in that region we have $p_r(x) - \gamma p_g(x) < 0$ and $D(x) \geqslant 0$. Hence, we obtain that

$$\underset{x \sim \mathbb{P}_r}{\mathbb{E}} [D(x)] - \gamma \underset{x \sim \mathbb{P}_g}{\mathbb{E}} [D(x)] \leqslant \int_{p_r(x) \geqslant \gamma p_g(x)} (p_r(x) - \gamma p_g(x)) dx + 0 = E_\gamma(\mathbb{P}_r \parallel \mathbb{P}_g)$$

This shows (3) and makes it clear where the choice of optimal function (4) came from. $\qquad\square$

