# OpenReview forum: "Hockey-Stick GAN"
_ICLR.cc/2018/Workshop — Accept_

### Official Review · AnonReviewer3 · 2018-03-09
**Overall good but needs more experiments**

**Rating:** 7
**Confidence:** 4

**Review:**

This paper proposes a new objective for GAN by the hockey-stick divergence. It has solid theoretical results.  Pros：
1. Utilizing the hockey-stick divergence to generate a new objective shows promising results in dealing with mode collapse and unstable convergence in GAN. The idea is novel.
2. This paper is well written and easy to follow.

Cons：
1. The only experiment is conducted on synthetic Gaussian data with shallow neural networks. The authors may want to conduct experiments on large-scale real-world data.
2. The hockey-stick divergence can equal zero even when the variables are different，which may lead to potential problems.

---

### Official Review · AnonReviewer1 · 2018-03-12
**Interesting special case of f-GANs**

**Rating:** 6
**Confidence:** 4

**Review:**

The paper investigates hockey-stick divergences for GANs. Hockey-stick divergences are a special case of f-divergences, which have been investigated for GANs in f-GAN work by Nowozin et al.

The paper discusses how different values of gamma recover other divergences. Since f-GANs can also recover other divergences through choice of f, I'd like to see more discussion of when/why hockey-stick GANs are better than f-GANS.

Would it be possible to use different gamma values for training discriminator and generator?
This idea has been proposed for f-GANs by Poole et al 2017 as well as "Learning in Implicit Generative Models" https://arxiv.org/abs/1610.03483

The paper presents experiments on only synthetic experiments. Would be great to include experiments on additional datasets and report other evaluation metrics as well.

---

### Official Review · AnonReviewer2 · 2018-03-12
**Very interesting look at a family of f-divergences with GANs**

**Rating:** 7
**Confidence:** 3

**Review:**

So, I like this direction. This is a pretty interesting application of a family of f-divergences (the hockey-stick divergences) as well as the necessary theory. It would be interesting to see if adjusting gamma can be used to slowly temper the model towards TV. Also, I have questions about the gamma parameter and optimality / non-optimality (it's unclear how optimal the discriminator was in the experiments), since it seems like GANs do well in the non-optimal case (regularization, early-stopping on the discriminator, etc). Finally, the results are counter-intuitive, no? It's stated that gamma > 1 can measure 0 divergence for P != 0, but it's exactly the gamma > 1 experiment that does best (correct me if I'm wrong). Is there some reason for this?

---

### Decision · Program_Chairs · 2018-03-20
**ICLR 2018 Workshop Acceptance Decision**

**Decision:**

Accept

**Comment:**

Congratulations, your paper was accepted to the ICLR workshop.